# The Role of CXCL4 in Systemic Sclerosis: DAMP, Auto-Antigen and Biomarker

**DOI:** 10.3390/ijms26062421

**Published:** 2025-03-07

**Authors:** Silvia Porreca, Anna Mennella, Loredana Frasca

**Affiliations:** National Center for Global Health, Istituto Superiore di Sanità, 00161 Rome, Italy; silvia.porreca@iss.it (S.P.); anna.mennella@iss.it (A.M.)

**Keywords:** autoimmunity, systemic sclerosis, CXCL4

## Abstract

Systemic sclerosis (SSc) is an autoimmune disease characterized by specific autoantibodies, vasculopathy and fibrosis of the skin and internal organs. In SSc, chronic activation of the immune system is largely sustained by endogenous inflammatory mediators that act as damage-associated molecular patterns (DAMPs), which activate Toll-like receptors (TLRs). Major autoantigens are nucleic acids or molecules that are able to bind nucleic acids. It is important to identify solid and predictive biomarkers of both disease activity and disease subtype. CXCL4 has been regarded as a new biomarker for early SSc in recent years, and here, we discuss its modulation over the course of a disease and after pharmacological interventions. Moreover, we provide evidence that CXCL4, in addition to being a biomarker of SSc subtypes and a prognostic marker of disease severity, has a dual pathogenic role in SSc: on the one hand, in complex with self-nucleic acids, CXCL4 acts as a DAMP for IFN-I and pro-inflammatory cytokines’ release by innate immune cells (such as dendritic cells); on the other hand, CXCL4 is a target of both antibodies and T cells, functioning as an autoantigen. CXCL4 is certainly an interesting molecule in inflammation and autoimmunity, not only in SSc, and it may also be considered as a therapy target.

## 1. Introduction

Autoimmune diseases are characterized by aberrant activation of both the innate and adaptive immune system, with disruption of the mechanisms preventing self-antigens’ recognition. The objective of this review is to explore the role of CXCL4, a molecule with pleiotropic functions, as a damage-associated molecular pattern (DAMP) with the ability to stimulate Toll-like receptors (TLRs), as an autoantigen and as an important biomarker in systemic sclerosis.

## 2. Systemic Sclerosis (SSc) Is an Autoimmune Disease

Systemic sclerosis (SSc), also referred to as scleroderma, corresponds to the definition of an autoimmune disease due to the presence of specific autoantibodies, in addition to vasculopathy and fibrosis of the skin and internal organs. SSc is an autoimmune disease with the highest mortality among rheumatic diseases [1]. Damages to the endothelium are believed to cause chronic tissue inflammation, resulting in the release of damage-associated molecular patterns (DAMPs), which activate the innate immune system [2]. The latter secondarily activate the adaptive immune system, thus leading to a further amplification of the inflammation [3,4,5]. 

### 2.1. SSc Classification, Diagnosis and Manifestations

Based on the extent of skin damage, two forms of the disease can be distinguished: limited cutaneous SSc (lcSSc), characterized by fibrosis at the elbows, knees, face and neck, and diffuse cutaneous SSc (dcSSc), involving skin distal and proximal to the knees. This is the most severe form, which is associated with fibrosis of the internal organs. Other manifestations of SSc include intestinal disease, pulmonary arterial hypertension (PAH), inflammatory arthritis, calcinosis and cardiomyopathy [3,6]. SSc is much more frequent in women than in men, with a ratio of (9:1) [7,8]. The trigger of SSc is unclear and involves the interaction between genetic and environmental factors. The earliest manifestation of SSc is Raynaud’s phenomenon (RP), a peripheral vasoconstriction combined with endothelial damage and the presence of SSc-specific autoantibodies. The most specific autoantibodies are anti-topoisomerase (ATA), anti-centromere (ACA) and anti-RNA polymerase III antibodies. The third important clinical parameter for SSc diagnosis is the alteration of capillaries, which can be detected by nailfold capillaroscopy [3,4,5]. One of the typical features of SSc is the activation of platelets, which contain several molecules with pro-fibrotic properties in their granules. Endothelial dysfunction may result in platelet activation and subsequent release of pro-fibrotic mediators that can stimulate fibroblasts to produce excessive amounts of collagen with the occurrence of fibrosis, a process not properly regulated in SSc [9].

### 2.2. Immunological Actors in SSc

Immune dysregulation in SSc involves the activation and recruitment of several immune cells associated with the production of autoantibodies and cytokines, such as interleukin (IL)-1, IL-4, IL-6 and IL-13, and transforming growth factor (TGF)-β. Moreover, SSc is a disease with frequent Type I interferon (IFN-I) signature, present in half of the patients [10]. IL-6 and TGF-β are known to be crucial in the pathogenesis of SSc. TGF-β is a pro-fibrotic factor, as it activates fibroblasts, stimulates collagen synthesis and promotes myofibroblast differentiation. Increased levels of IL-6, which can be found in the sera and skin of SSc patients, correlate with disease severity, in particular with the extent of skin and pulmonary involvement, and they can also predict SSc-linked interstitial lung disease (ILD), decline and mortality [11]. Among the factors overexpressed in blood and tissues, there is also C-X-C motif ligand 4 (CXCL4), and although platelets are the main producers of CXCL4, the pDCs from SSc patients were found to secrete CXCL4 [12,13].

Upon activation, pDCs produce IFN-I, which creates an inflammatory environment in the tissues. This inflammation is linked to the aberrant expression of TLR8 by pDCs in SSc patients [14]. This also contributes to disease progression, since signaling through TLR8 induces the production of CXCL4, which creates a pathogenic loop. Additionally, TLR8 expression leads to an increased infiltration of pDCs into the tissues, exacerbating the disease and resulting in worse skin fibrosis [15]. Of note, another chemokine, C-X-C motif chemokine ligand 10 (CXCL10), also known as Interferon-γ-induced protein 10 (IP-10), is highly expressed in SSc. CXCL10 is a major pro-inflammatory Th (T helper)-1 cell (Th1) chemokine involved in the pathophysiology of multiple autoimmune diseases [16]. It is released in response to IFN-γ by CD4 and CD8 T cells and by natural killer (NK) cells. CXCL10 works via binding to C-X-C chemokine receptor 3 (CXCR3) and attracts inflammatory cells, including monocytes, Th1, CD8 T cells, NK, NKT and dendritic cells [16]. CXCL10 protein levels are elevated systemically (serum) and in the local (lung) compartments of SSc patients with ILD compared to SSc patients without ILD. Furthermore, high systemic levels of CXCL10 may be associated with a higher risk of a new onset of ILD in patients with SSc [17].

## 3. The Important Role of Toll-like Receptors in SSc: CXCL4 as a DAMP

Aberrant activation of Toll-like receptors (TLRs) is central in the pathogenesis of SSc. TLRs are receptors for microbial antigens, which are referred to as pathogen-associated molecular patterns (PAMPs), but they also recognize endogenous signals of inflammation, which are released under stress conditions or after trauma (DAMPs, damage-associated molecular patterns). Stimulation of TLRs by PAMPs and DAMPs activates both immune and non-immune cells, among the latter, fibroblasts and endothelial cells, both important in SSc pathogenesis and exhibiting dysfunctionality in repairing wounds [4]. It is believed that SSc begins with vascular damage, which leads to the activation and recruitment of innate immune cells, which initiate the response leading to chronic inflammation, increased vascular damage and abnormal wound repair. The inflammatory environment also leads to the activation and dysregulation of fibroblasts, resulting in the excessive production of extracellular matrix (ECM) components and fibrosis. This fibrosis further promotes the recruitment and activation of immune cells, creating a feedback loop. Therefore, the current treatment strategies include the use of immunosuppressive and anti-fibrotic agents [8]. It is via TLRs that inflammation becomes chronic in SSc, and several molecules can act as DAMPs in the disease [4,5]. Among these DAMPs, there is C-X-C motif ligand 4 (CXCL4) [18,19,20,21]. Compared to CXCL10, CXCL4 possesses some features and functions that render the molecule an important DAMP and an IFN-I inducer.

### 3.1. CXCL4: Structure and Function

CXCL4 is a 70-amino-acid chemokine belonging to the CXC family of chemokines. The monomeric form is characterized by a hydrophobic sequence with three antiparallel beta sheets and an amphipathic alpha helix in the carboxy-terminal portion. Its tetrameric form includes two asymmetric dimers and forms a cylindrical structure with a positively charged equatorial ring [22]. CXCL4 (and, to a lesser extent, CXCL4-L1, the non-allelic CXCL4 variant) belongs to the category of antimicrobial peptides, the cationic and amphipathic effectors of innate immunity that carry out their microbicidal actions via a broad spectrum of mechanisms, including destabilization of the pathogen’s membrane. Specifically, CXCL4 belongs to the kinocidins family, chemokines that represent a natural antibiotic in tissues, which are released directly into the bloodstream following trauma or infection [23].

### 3.2. Expression of CXCL4 in Cells and Its Receptors

CXCL4 is synthesized in megakaryocytes and then accumulates in platelet granules, bound to two molecules of chondroitin sulfate and released in response to protein kinase C. It is also present in plasma, mast cells, and it is released by monocytes and T cells. It is highly produced by pDCs in SSc patients [24]. CXCL4 possesses a high affinity for glycosaminoglycans (GAGs). One study showed that CXCL4, by binding to GAG chains expressed within the vascular extracellular matrix, is able to mediate leukocyte recruitment in response to inflammatory stimuli [25]. However, CXCL4 also produces an anticoagulant action by stimulating the generation of active protein C through binding to thrombomodulin [26]. Among the GAG molecules, CXCL4 can bind heparin, an anticoagulant released from mast cell granules, which binds to antithrombin, causing its conformational change, a process that increases its affinity for thrombin. This also leads to inhibition of the coagulation cascade [20]. CXC chemokine receptor 3 (CXCR3) is a signal receptor that has been associated with the functions of CXCL4 (albeit at high concentrations of CXCL4), which is present in two isoforms: CXCR3-A, present on leukocytes, which mediates cell proliferation, survival and migration, and CXCR3-B, expressed on endothelial cells, which mediates apoptosis and inhibition of cell proliferation [24,25,26,27]. In recent years, a new family of chemokine receptors, defined as "atypical chemokine receptors" (ACKRs), has been identified; they differ from classical chemokine receptors in their inability to induce G-protein-mediated signaling. Unlike canonical chemokine receptors, which are expressed on immune cells, those receptors are mostly found on endothelial cells and barrier organ epithelial cells. The different cellular effects and expression patterns have led to speculation that CXCR3-B may act as an ACKR and inhibit cell migration and proliferation [28]. It has been proposed that CXCR3-B might mediate the anti-angiogenic activity of CXCL4, exerting an inhibitory effect on endothelial cells [26]. Interestingly, CXCL4 also plays a role in the development of atherosclerosis; in fact, it is able to promote the differentiation of monocytes into macrophages, cells that participate in the formation of atherosclerotic plaque. Moreover, in vitro experiments showed that CXCL4 binds to oxidized LDL, mediating their uptake by macrophages, thus promoting the formation of foamy cells [29]. CXCL4 is a chemokine with pleiotropic functions (Figure 1). The modular determinants from its structural components act independently and govern the antimicrobial functions of CXCL4 against specific pathogens, e.g., Staphylococcus aureus, Salmonella typhimurium and Candida albicans strains [23]. CXCL4 is described to have anti-infectious properties, working as a HIV-1 inhibitor [30]. It can protect from lung pathogenesis caused by influenza (H1N1) respiratory infection [31]. Moreover, the TLR9 pathways in ANCA-associated vasculitis platelets enhance CXCL4 release and NETs formation and contribute to the pathogenesis of vasculitis [32]. CXCL4 exposure potentiates TLR3/7/8-driven polarization of human monocyte-derived dendritic cells (upregulation of CD83, CD86, MHC class I), increased secretion of IL-12 and TNF-α and increased stimulation of T cells [33]. CXCL4 can also induce intracellular calcium release and the migration of activated human T lymphocytes. In diseases such as atherosclerosis, following platelet activation, the CXCR3/CXCL4 axis has a role in T-cell recruitment and amplification of inflammation [34]. CXCL4 is a potent inhibitor of tumor-induced angiogenesis. CXCL4 might also act as a ligand for integrins, playing a crucial role in angiogenesis, especially by regulating early angiogenic steps, such as endothelial cell adhesion and cell migration [35]. In some individuals exposed to heparin, CXCL4/heparin complexes can be formed and may act as an immunogen, leading to the generation of CXCL4/heparin antibodies. This may lead to a severe clinical condition called heparin-induced thrombocytopenia (HIT), characterized by platelet activation and aggregation, thrombocytopenia and venous and arterial thrombosis [36]. CXCL4 also represents a broad activator of monocytes, inducing cell differentiation, survival, release of ROS and cytokines [37].

### 3.3. CXCL4 as a DAMP for IFN-I Amplification in pDCs

CXCL4 amplifies the response of the TLRs involved in IFN-I production, participating in the IFN-I signature in SSc. As we mentioned above, the structure of CXCL4 containing strongly clustered cationic charges and an amphipathic alpha helix at the carboxyl-terminal part is reminiscent of common structures typical of antimicrobial peptides [23]. CXCL4 has been shown to interact electrostatically with DNA by forming immune complexes that induce effective activation of pDCs. Indeed, CXCL4 has been shown to organize both self- or microbial DNA into a liquid–crystalline stimulatory structure that enables the internalization of complex-bound DNA. This DNA is subsequently transported into the endosomal compartment, where TLR3, TLR7/8/9 are positioned [38]. In endosomes, the DNA is "presented" in an appropriate manner to TLR9. We can therefore consider that, in this process, CXCL4 exerts a chaperon activity, protecting DNA from degradation and performing “adjuvant-like activity”. Not all DNA-binding molecules, however, can act as such “adjuvants". In order to do so, certain requirements are necessary: the peptide must be able to form aggregates of DNA molecules, which are ordered in crystal structures, which must be capable of interacting with TLR9 via multiple interactions, forming a cluster of receptors. This is the case for CXCL4 [38]. It is also necessary that a chaperon molecule protect the DNA from the endonucleases and exonucleases. Finally, for optimal TLR9 activation, an inter-DNA ligand distance of 3–4 nm is required [38,39,40] (Figure 2).

This phenomenon induces the hyperactivation of pDCs, with amplification of IFN-I production, and it may promote the breakdown of tolerance toward self-antigens [38]. PDCs can also be activated by CXCL4–RNA complexes, possibly by triggering TLR7 and TLR8 [41]. The DNA and RNA bound to CXCL4, forming a DAMP, can be derived from cells dying due to necrosis during inflammation but also from neutrophils undergoing a form of cell death called NETosis (neutrophil extra-cellular traps). Indeed, studies by others and our own data [38] indicate that NETosis could play a role in SSc [42].

A recent study shows that CXCL4 is able to retain CXCL4 in complex with CpGA (a DNA ligand) in the early endosome compartment, and this is responsible for the hyperproduction of IFN-I [14].

As mentioned above, CXCL10 is also highly present in SSc blood. It is interesting that CXCL10 was also shown to have the ability to bind self-DNA (human genomic DNA) and microbial DNA. However, CXCL10 was shown to be able to act as a TLR9 stimulator only in association with bacterial DNA, whereas in complex with self-DNA, its effect was poor [43]. 

### 3.4. CXCL4 Role as a DAMP in Stimulating Pro-Inflammatory Factors by Myeloid Dendritic Cells

As we reported in previous sections, CXCL4–DNA and CXCL4–RNA complexes have been observed in both circulation and dermis of patients with SSc, indicating that CXCL4 may indeed exert adjuvant activity in vivo [38].

It has also been shown that CXCL4, complexed with RNA, stimulates not only pDCs [4] but also myeloid dendritic cells (mDCs), especially if pretreated with IFN-I. MDCs are the main regulators of the immune response and express a wide range of receptors for PAMPS: TLRs, RIG-1-like receptors (RLRs) and NOD-like receptors (NLRs) [41,44]. Importantly, mDCs are the cells that present antigens in a "professional" manner, and they are crucial for triggering an adaptive immune response. It has already been published that CXCL4–RNA immune complexes amplify the maturation of IFN-I-pretreated mDCs in vitro [43]. Human monocyte-derived dendritic cells (MDDCs) are derived from peripheral blood monocytes in vitro, and they can be considered the in vitro counterpart of inflammatory monocytes that become DCs and localize to inflamed sites [44,45,46]. We demonstrated that CXCL4 complexed with self-RNA and CXCL4–RNA complexes stimulated the production of pro-inflammatory cytokines and some pro-fibrotic factors in MDDCs, including TNF-α, IL-12, IL-23, IL-8 and pro-collagen. The latter, together with TNF-α, plays a role in the development of pulmonary fibrosis and pulmonary arterial hypertension [47,48]. Notably, it has also been shown that CXCL4–RNA complexes circulate in SSc blood and correlate with both plasma IFN-I and TNF-α levels [44]. It is unclear what determinant enables the optimal stimulation of TLR8 (and perhaps TLR7) by CXCL4–RNA complexes. Figure 3 shows that CXCL4–DNA and CXCL4–RNA complexes can activate TLR7/8 and TLR9 in immune cells. Therefore, CXCL4 bound to DNA/RNA (self-derived but also of microbial origin) can act as a DAMP and contributes to inflammation in the same manner as other recognized DAMPs like Tenascin, fibronectin, S100A8 and S100A9 (alarmins) [4,5]. Anti-CXCL4 antibodies are also present in SSc (see below) and can concentrate CXCL4 for its internalization in immune cells.

### 3.5. CXCL4 Role as a DAMP in B-Cells

B cells express several TLRs [50]. B-cell activation requires three synergistically acting stimuli, which are the antigen-mediated B-cell receptor activation, stimulation via CD40 signaling and TLR triggering [51]. However, several groups have described human naïve B-cell activation via TLR stimulation only [52,53]. TLR7 and TLR9, and signaling via those molecules, can induce their activation and antibody production [54,55]. Using an in vitro model system of a mixture of memory and naïve B cells from buffy coats, we demonstrated that complexes formed by CXCL4 and self-DNA induce the production of high levels of IgG in B cells [41]. In the presence of IFN-I, CXCL4–RNA complexes were also effective in inducing the same effect [44]. Indeed, it has been shown that IFN-I increases TLR7 and results in polyclonal B-cell expansion, with differentiation toward Ig-producing plasma cells, an event that helps T cells and is independent of antigens [56]. As presented in Figure 4, B cells were stimulated by both CXCL4–DNA and CXCL4–RNA complexes, becoming antibody-secreting plasma cells. Thus, pDCs, which are stimulated to produce IFN-I in response to the same complexes, may concur with B-cell activation and plasma cell differentiation in vivo via IFN-I. This loop of cell activation can explain the high frequent anti-CXCL4 antibody generation in SSc (see below) [41,57].

## 4. CXCL4 as an Autoantigen

### 4.1. Autoimmunity in SSc

Autoimmunity in SSc is linked to inflammation, vasculopathy and fibrosis [60]. It is an important component of SSc, as autoreactive T cells and autoantibodies play a crucial role. Autoimmunity in SSc requires both innate and adaptive immune responses at humoral and cellular levels, which participate in disease initiation under the influence of specific genetic and environmental factors [60]. One characteristic feature of the disease is the production of a variety of autoantibodies to nuclear antigens. There are some examples of typical SSc autoantibodies that activate innate immune cells, fuel inflammation and can contribute to IFN-I signature and fibrosis [61]. For instance, Kim’s group demonstrated that SSc sera containing autoantibodies, such as anti-centromere (ACA) and anti-topoisomerase (ATA) antibodies, induced high levels of IFN-α in healthy donor (HD) peripheral blood mononuclear cells (PBMCs) in a pDC-dependent manner [61]. It has been shown that 95% of patients with SSc have circulating antinuclear autoantibodies (AAbs) in blood. These AAbs are a serological hallmark of SSc and are used as biomarkers for establishing an early and accurate diagnosis [62].

### 4.2. CXCL4 Is Recognized by Antibodies and T-Cells in SSc

It has been demonstrated that CXCL4 represents a new autoantigen in SSc. Our group showed that an IgG response to CXCL4 characterizes SSc. Anti-CXCL4 autoantibodies are significantly higher in SSc patients with active disease and with lung fibrosis [9] compared to patients without lung fibrosis. One of the effector functions of these anti-CXCL4 autoantibodies could be to concentrate and more efficiently deliver CXCL4–DNA/RNA complexes to immune cells, such as pDCs, to further amplify IFN-α production in SSc and the secretion of inflammatory factors, such as TNF-α or IL-12/IL-23 [41]. Anti-CXCL4 antibodies were higher not only in patients with pulmonary fibrosis but also in those with digital ulcers; moreover, they were associated with disease activity and correlated with IFN-α present in blood [57]. It has been shown that CD4 T cells proliferated to CXCL4, and this proliferative capacity significantly correlated with the magnitude of antibody reactivity to CXCL4 in a patient cohort. This observation indicated that CXCL4 may act as a novel autoantigen in SSc, activating both humoral and cellular responses [41], as presented in Figure 5. Growing evidence indicates that IL-17 from Th17 cells plays a key role in various autoimmune and chronic inflammatory diseases and is associated not only with T-cell-mediated tissue injury but also with the production of pathogenic autoantibodies [63]. Regarding this aspect, a study has shown that CXCL4 is able to skew T-helper responses toward a Th17 phenotype [64]. In addition, CXCL4 was previously shown to skew CD4 T-cell responses toward a Th2 phenotype, although induction of Th17 cytokines was not addressed at that time [65]. 

It is well known that CXCL4 is a heparin-binding protein [66]. Complexes formed by CXCL4 and heparin play a role in a rare disease called heparin-induced thrombocytopenia (HIT), in which CXCL4 bound to heparin becomes the target of anti-CXCL4 antibodies with pathogenic activity. These antibodies determine platelet activation and thrombosis [36,67]. Neutrophil activation by CXCL4–anti-CXCL4 complexes determines an enhanced cell adhesion via selectins and integrins. This can be relevant for HIT patients, since activated neutrophils, adherent on the vessel wall, may damage tissue and initiate the process of thrombosis and vasculitis, with NETosis also playing a role in this process [68,69]. HIT is a severe condition in which heparin-dependent anti-CXCL4 antibodies are generated, which exert pathogenic functions by targeting CXCL4 attached to endogenous heparan-sulfate on endothelial cells, causing injury to these cells. It has been demonstrated that SSc patients have a significantly higher concentration of heparin-dependent anti-CXCL4 antibodies as compared to healthy donors; in particular, antibodies to CXCL4 can be either heparin-dependent or heparin-independent in SSc [67]. We showed that SSc patients who do not have significant amounts of anti-CXCL4 antibodies usually have HIT antibodies, as if the two types of antibodies were mutually exclusive, although this finding needs to be confirmed in larger SSc cohorts [67]. One explanation for this mutually exclusive expression can be found in a paper by Sachais et al. [70], in which the authors showed that some HIT antibodies also bind to CXCL4 alone but with much lower affinity than to CXCL4–heparin complexes. Antibodies with such behavior were shown to induce the clustering of CXCL4. They acted like heparin, inducing CXCL4 oligomerization, which increases immunogenicity. This action promotes the generation of further epitopes by cross-linking CXCL4 tetramers. Given the high likelihood of generating anti-CXCL4 antibodies in SSc, as CXCL4 is over-expressed, this mechanism could be frequently operative in SSc and favor the production of HIT antibodies. 

### 4.3. CXCL4 Can Favor Loss of Immune Tolerance

CXCL4 can complex with DNA and form stable nanoparticles with DNA, in part due to the cationic nature of CXCL4, which leads to the TLR9 response in pDCs [20]. It has been demonstrated that innate immune recognition via TLR9 is required for the establishment of B-cell central tolerance [46]. Cakan et al. showed that CXCL4 inhibits TLR9 function in B cells by increasing TLR9 ligand cellular uptake, preventing its delivery to late endosomes/lysosomes where TLR9 resides, thereby hindering the sensing of dsDNA by TLR9 and abrogating TLR9 function [46]. This leads to impairments in central B-cell tolerance through failure to properly upregulate activation-induced cytidine deaminase (AID), which is required for the selection of developing autoreactive B cells in bone marrow [47]. CXCL4-induced TLR9 impairment in B cells from SSc patients also likely results in the secretion of autoantibodies targeting topoisomerase I or centromere proteins, both of which bind dsDNA in patients with diffuse or limited SSc, respectively. Therefore, autoimmune diseases characterized by serum autoantibodies that target self-antigens that interact with TLR9 ligand dsDNA likely result from impaired TLR9 function in B cells [46]. These data therefore provide a rationale for the development of novel therapeutic strategies for SSc and SLE that will aim to restore TLR9 function in B cells to maintain both central and peripheral B-cell tolerance and suppress autoimmune manifestations. Finally, the increased proportions of activated platelets in SSc and pDCs may represent important systemic and tissue-specific sources of CXCL4 that will alter TLR9 responses in both B cells and pDCs [71]. In our study, CXCL4–DNA/RNA complexes stimulated memory B cells via TLR7/9 (as described above), which became antibody-secreting plasma cells. The mechanisms at work for memory and naïve B cells and B-cell precursors could be different, and the same TLRs can participate in different functions at different stages of B-cell differentiation.

## 5. CXCL4 as a Biomarker in SSc

### 5.1. Role of CXCL4 as a Biomarker in SSc

A large, multicenter proteomics study, published in *N. Eng. J. Med* in 2014, identified CXCL4 as a biomarker of SSc, particularly early SSc. It was observed that elevated CXCL4 levels in the blood and tissues of patients with SSc not only correlated with skin and lung damage but were also predictive markers of SSc progression to more aggressive forms [24]. CXCL4 expression has been shown to positively correlate with IFN-I levels in the plasma of SSc patients, suggesting an important role of CXCL4 in the IFN-I signature [38]. One of the ways in which CXCL4 can promote IFN-I signature is the ability to bind nucleic acids and activate TLR9 and TLR7 [4], as above. Other biomarker chemokines, in addition to CXCL4, have been associated with interstitial lung disease (ILD), which develops in 65–85% of SSc patients and is characterized by endothelial dysfunction, extracellular matrix deposition and subsequent pulmonary fibrosis [72]. It has also been shown that a decrease in CXCL4 plasma levels is associated with improvements in lung function in patients treated with immunosuppressive therapies associated with ILD. In fact, one study described that 142 patients who had received either Cyclophosphamide for one year or Mycophenolate for two years showed a decrease in circulating CXCL4 levels, accompanied by an improvement in lung function after twenty-four months. These results suggest that medium-term changes in CXCL4 may have predictive significance for long-term progression to lung disease in SSc patients undergoing immunosuppressive therapy [73]. 

### 5.2. CXCL4 Compared to CXCL10 as Biomarker

As mentioned previously, a chemokine similar to CXCL4, CXCL10, was found to be highly involved in SSc [16]. CXCL10, which shares the receptor CXCR3 with CXCL4 [71], has been associated with bacterial DNA–induced IFN-I production and enhancement of wound healing [43].

Normally, this chemokine is produced by a wide range of cell types, including monocytes, neutrophils, endothelial cells, keratinocytes, fibroblasts, mesenchymal cells, dendritic cells, hepatocytes and astrocytes [74]. Higher CXCL10 is not only associated with a more severe disease prognosis [75] but, remarkably, precedes the development of definite SSc from a preclinical condition of undifferentiated connective tissue disease (UCTD) at risk of SSc, so far referred to as very early diagnosis of systemic sclerosis (VEDOSS) [76]. It has been demonstrated that the prostacyclin analog iloprost acts by binding to the prostacyclin receptor (IP receptor), causing the elevation of cyclic AMP (cAMP) [77], which can cause a reduction in CXCL10 levels, inhibiting CXCL10 secretion in activated human endothelial cells and dermal fibroblasts, thus preventing activation of the paths synergistically involved in chemokine release. Indeed, this chemokine is known to play a pivotal role at the onset of vascular and immune perturbation, likely acting as an early mediator toward fibrosis, within the cross-talk among endothelium/fibroblasts/immune systems [78]. Interestingly, iloprost was found to also downregulate CXCL4 expression in the blood of SSc patients with early disease, which was followed by a decrease in IFN-I levels too [79]. The mechanisms underlying this effect have not been explored. One possible explanation could be the capacity of prostacyclin analogs, such as iloprost, to block NETosis at the blood vessel level, as suggested by a recent study [80]. The finding that iloprost, in addition to the well-known vasodilator effect, can counteract CXCL10 release by human endothelial cells and fibroblasts and also downmodulate CXCL4 in early SSc might open new scenarios for early treatments in SSc.

## 6. CXCL4 as a Therapy Target

One way of blocking the side effects of CXCL4 is to inhibit its production. It has been demonstrated that hypoxia and stimulation of TLR9 are the factors leading to increased CXCL4 production in plasmacytoid dendritic cells via increased production of reactive oxygen species from mitochondria (mtROS) [81]. Hypoxia is often considered the driving force behind many pathological hallmarks of SSc [82]. The hypoxic environment makes pDCs more prone to respond to viral or bacterial infection or endogenous ligands via TLR signaling, culminating in high production of CXCL4 and acceleration of disease progression. Furthermore, CXCL4 promotes the genetic imprinting of DCs, making them more prone to TLR stimulation, and changes the DCs in pro-fibrotic cells via direct matrix production, as well as indirectly, by inducing myofibroblast transition [33]. This suggests that CXCL4 plays an essential role in the circle of inflammation, hypoxia and fibrosis, as observed in SSc. Interestingly, blocking mtROS with a specific mtROS inhibitor (mitoQ) significantly reduced the production of CXCL4 in pDCs. Hence, there seems to exist a link between hypoxia and disease progression orchestrated by CXCL4—an observation with great therapeutic relevance [81].

Our group demonstrated that CXCL4 forms nano-crystalline complexes with DNA, which enables self-DNA to induce immune amplification via TLR9 activation [38], and that CXCL4 also works as a “danger signal” in complex with RNA [44]. It has been shown that iloprost, a synthetic prostaglandin and one of the most used drugs in the treatment of vascular manifestation of SSc [83], is able to dose-dependently inhibit IFN-α induced by CXCL4–DNA and CXCL4–RNA complex stimulation in pDCs. This suggests that iloprost may be useful in decreasing the two important interconnected factors: CXCL4 and IFN-I [74]. Notably, activation of the IFN-I pathway as an early event in SSc is associated, together with high CXCL4, with more severe disease manifestation and poor prognosis [24]. Furthermore, this study demonstrated that iloprost treatment reduced disease activity in an SSc subgroup with a shorter disease duration. Therefore, this suggests that early patients may be those who could better benefit from a timely use of iloprost to block CXCL4 activity (in complex with nucleic acids) and IFN-I-linked pathways. Early treatment may translate into a slower progression of the disease and fewer complications thereafter [79].

## 7. Conclusions

In this article, we tried to offer a comprehensive review regarding the role of CXCL4 in SSc by offering a wider point of view than that found in other literature works, as we wanted to touch on all aspects relative to the functions of CXCL4 in SSc. CXCL4 is a molecule that appears to participate in several pathogenic mechanisms of the disease, and it is not only a biomarker or an IFN-I inducer. CXCL4 may affect several cell types of the immune system, including B cells and their secretion of autoantibodies, as well as T cells. The adaptive immune system appears to be stimulated to react to CXCL4 as an autoantigen, a phenomenon that is likely favored by the “adjuvant-like activity” (DAMP) of this molecule. We provide a new perspective to stimulate other studies in the field of SSc and CXCL4. These future perspectives for CXCL4 research in SSc should be focused on a better understanding of how CXCL4 is modulated during pharmacological interventions to understand whether it can be used as a marker to predict therapy success in clinical settings. Moreover, a focus on CXCL4 as the target of therapy itself in SSc and perhaps in other chronic inflammatory diseases warrants future investigations.

## Figures and Tables

**Figure 1 ijms-26-02421-f001:**
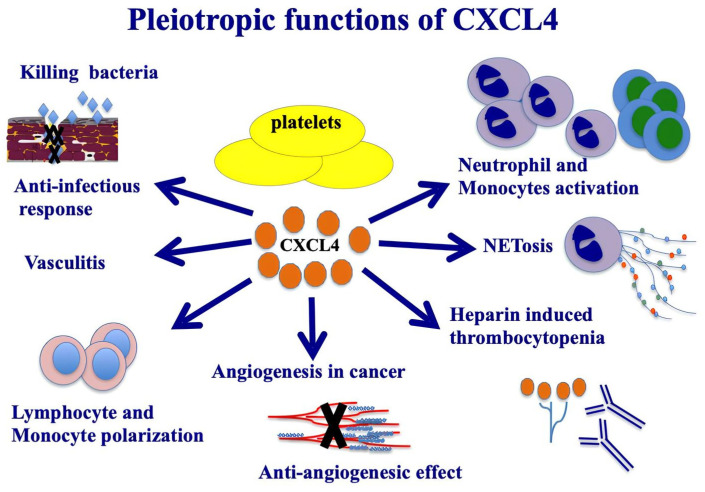
Pleiotropic functions of CXCL4. CXCL4 is a member of kinocidins, with direct capacity for bacterial killing. The image illustrates the most important functions of CXCL4, which are also relevant in autoimmunity and in SSc.

**Figure 2 ijms-26-02421-f002:**
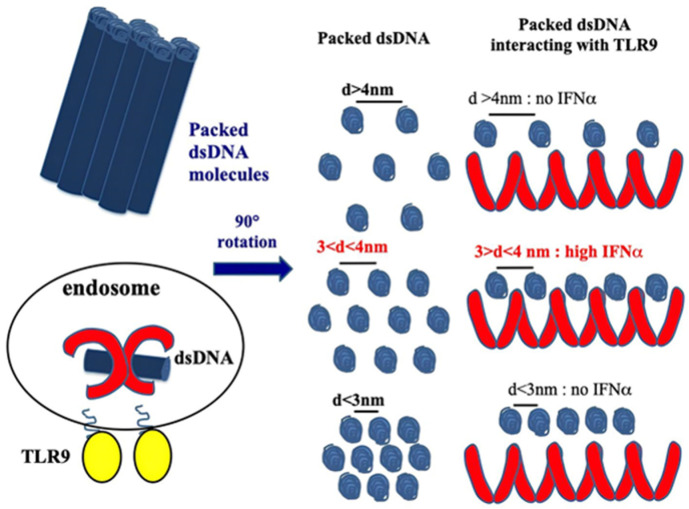
CXCL4 organizes dsDNA into immunogenic liquid–crystalline complexes suitable for TLR9-mediated IFN-α production. Inter-DNA spacing close to the steric size of TLR9 (d = 3–4 nm) allows the optimal binding of columnar DNA lattices to TLR9-clustered arrays. CXCL4 organizes DNA into liquid–crystalline columnar lattices at an inter-DNA spacing compatible with TLR9 amplification. DNA fragmentation increases the total number of discrete DNA fragments for the same mass of DNA, with optimal close packing of DNA ligands [38].

**Figure 3 ijms-26-02421-f003:**
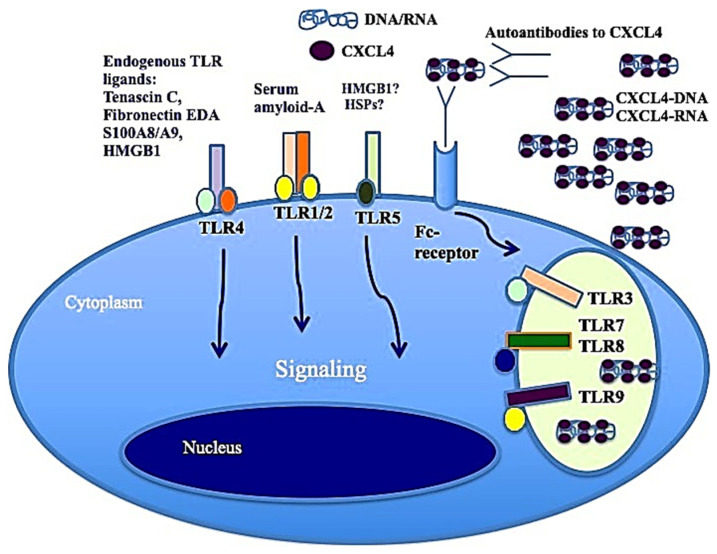
CXCL4–DNA and CXCL4–RNA complexes activate immune cells via TLRs. As with other well-known DAMPs like Tenascin-C, acting via TLR4 [49], fibronectin, S100A8 and S100A9 (alarmins), when CXCL4 binds to DNA/RNA, it acts as a DAMP and participates in the inflammation process.

**Figure 4 ijms-26-02421-f004:**
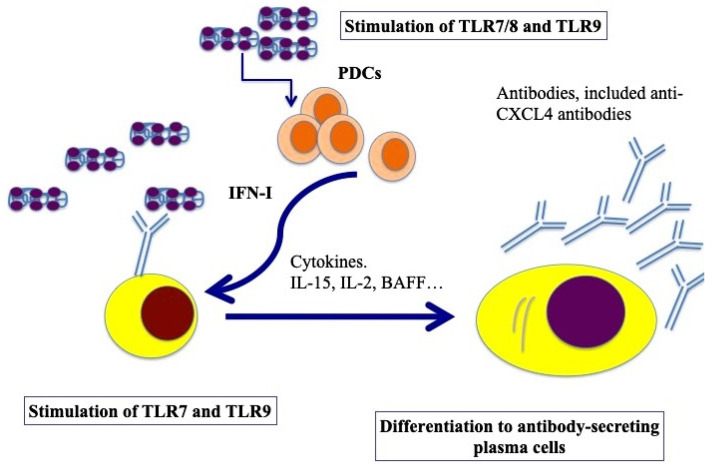
Model for B-cell activation via CXCL4–DNA/RNA complexes. After stimulation through TLR7/8/9 by CXCL4–DNA/RNA complexes, pDCs become activated and start to produce IFN-I, which, together with BAFF, IL-15, IL-2, help plasma cell differentiation (pDCs can also produce BAFF, [58], whereas IL-15 and IL-2 can be produced by T cells but also by stroll cells, monocytes, etc. [59]). Afterward, B-cell transition and differentiation in antibody-secreting plasma cells occurs. In SSc, activated B cells start to produce antibodies, among which there could be anti-CXCL4 antibodies.

**Figure 5 ijms-26-02421-f005:**
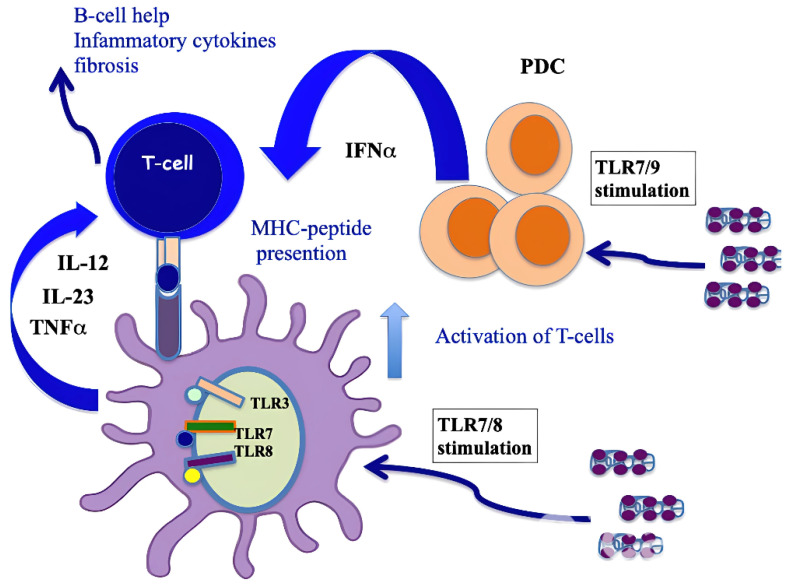
CXCL4 as an autoantigen in SSc. When complexed with nucleic acids, CXCL4 stimulates pDCs and mDCs via TLRs, which activates both cell types and induces the production of IFN-I, IL-12, IL-23 and TNF-α. IFN-I released by pDCs implements the antigen-presenting cell capacity of mDCs, whereas pro-inflammatory cytokines can activate CD4 T cells (including Th17 cells and perhaps CD8 T cells [64]). T cells specific to CXCL4 are likely to provide B-cell help for autoantibody production [41] or directly induce an inflammation that favor fibrosis. An algorithm that predicts the binding capacity of CXCL4-derived epitopes to HLA molecules showed that the sequence of CXCL4 possesses “binding motifs” for several HLA-DR alleles represented in Caucasians (DR4, DR1 and DR11), which may explain the recognition by CD4 T cells [41].

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
