# Peer review of "The Role of CXCL4 in Systemic Sclerosis: DAMP, Auto-Antigen and Biomarker"

_ijms, 2025, doi:10.3390/ijms26062421_

Round 1

Reviewer 1 Report

Comments and Suggestions for Authors

The paper is interesting and well written. The authors discuss the role of CXCL4 as both DAMP and auto-antigen in SSc, and how these two functions can be linked. I suggest to discuss the interaction between CXCL4  and Th17 cells that play a role in chronic inflammatory immune-mediated diseases (see and add as reference the paper by Murdaca et al concerning Th17 cells in chronic inflammatory immune-mediated diseases).

Comments on the Quality of English Language

Minor english editing

Author Response

REVIEWER 1

The paper is interesting and well written. The authors discuss the role of CXCL4 as both DAMP and auto-antigen in SSc, and how these two functions can be linked. I suggest to discuss the interaction between CXCL4 and Th17 cells that play a role in chronic inflammatory immune-mediated diseases (see and add as reference the paper by Murdaca et al concerning Th17 cells in chronic inflammatory immune-mediated diseases).

We thank the reviewer for the appreciation of our work. We have added the reference indicated and two more references that address the CXCL4 polarizing capacity of CXCL4 on T helper cells.

Reviewer 2 Report

Comments and Suggestions for Authors

The present review comprehensively describes the major roles played by CXCL4 in SSc.

I found it clear and well written.

I only have few minor concerns to be addressed.

Paragraph 2.1 - "REynaud's phenomenon" - Please, correct the mistyping with RAynaud's phenomenon.

Paragraph 3.3 -"CXCL4 has been shown to interact electrostatically with the DNA by forming immune complexes that induce effective activation of pDCs" - Please, clarify what's the origin of the extracellular DNA bound by CXCL4.

Paragraph 3.3 - "In order do so" - Please, correct the mistyping with "in order to do so".

Several concepts are discussed in different paragraphs of the review, thus making the reading more difficult. For example, CXCL4 as heparin-binding protein, inducing HIT disease, in paragraphs 3.2 and 4.2. Wherever possible, I would suggest to avoid this fragmentation, introducing each concept only once in a single part of the manuscript. Please, revise the manuscript accordingly.

Figure 5 is not clearly comprehensible. Recognition of CXCL4 by T cells is not clearly shown. Legend does not properly explain what shown in the figure.  

Author Response

REVIEWER 2

The present review comprehensively describes the major roles played by CXCL4 in SSc.

I found it clear and well written.

I only have few minor concerns to be addressed.

Paragraph 2.1 - "REynaud's phenomenon" - Please, correct the mistyping with RAynaud's phenomenon.

We thank the reviewer for the positive evaluation of our manuscript. We corrected the mistake.

Paragraph 3.3 -"CXCL4 has been shown to interact electrostatically with the DNA by forming immune complexes that induce effective activation of pDCs" - Please, clarify what's the origin of the extracellular DNA bound by CXCL4.

We thank the reviewer for this interesting question, which needs clarification. We have added some statements in the the paragraph and one more reference.

Paragraph 3.3 - "In order do so" - Please, correct the mistyping with "in order to do so".

We thank, we have corrected

Several concepts are discussed in different paragraphs of the review, thus making the reading more difficult. For example, CXCL4 as heparin-binding protein, inducing HIT disease, in paragraphs 3.2 and 4.2. Wherever possible, I would suggest to avoid this fragmentation, introducing each concept only once in a single part of the manuscript. Please, revise the manuscript accordingly.

We agree with the reviewer and we have tried to render clearer some parts of the text.

Figure 5 is not clearly comprehensible. Recognition of CXCL4 by T cells is not clearly shown. Legend does not properly explain what shown in the figure.  

We have improved the Figure legend, to explain better the mechanisms depicted in the Figure.

Reviewer 3 Report

Comments and Suggestions for Authors

In this review, th authors provide information on the role of CXCL4 in systemic sclerosis and the different processes by which CXCL4 can promote disease progression, as well as its utility as biomarker or therapy target.

The manuscript is very well written and provides a comprehensive review on the topic.

The manuscript provides detailed information on the role of CXCL4 in SSc. It provides valuable insights on CXCL4 participation in the progression of the disease, its role as an autoantigen itself and as a biomarker to monitor patients' response to SSC therapy. The topic is relevant to the field, not only for SSc but in other autoimmune diseases. Likewise, it is an original topic as this review concentrates only on CXCL4, which participates in several pathogenic mechanisms of the disease, offering a wider view than other reviews on the field that mention the participation of CXCL4 from individual points of view (e.g., pDC, autoantibodies origin, dendritic cells).  Being a narrative review, no specific gap in the field is covered, but it may help to provide a new perspective to other studies in the field of SSc. Likewise, being a narrative review, a specific methodology was not followed, although the authors could provide more information about studies that found contradictory results and discuss them; the conclusions are a summary of the main points reviewed.

Regarding the references, they are appropriate for this kind of manuscript.

The manuscript includes three figures appropriate for the information being reviewed, although its quality could be improved.

 Some very minor points should be addressed.

The title of the submission is different from the title of the manuscript. Please use the title of the manuscript, since the title of the submission do not appropiately reflect the focus of the manuscript.

Some minor issues in punctuation need correction (e.g. point at beginning of paragraph, page 6 line 9). 

Correct "as autoreactive T-cells and autoantibodies play as crucial role in the disease" (page 8).

On page 3, line 8, it seems that the phase was left unfinished just before reference [6].

Author Response

REVIEWER 3

In this review, the authors provide information on the role of CXCL4 in systemic sclerosis and the different processes by which CXCL4 can promote disease progression, as well as its utility as biomarker or therapy target.

The manuscript is very well written and provides a comprehensive review on the topic.

The manuscript provides detailed information on the role of CXCL4 in SSc. It provides valuable insights on CXCL4 participation in the progression of the disease, its role as an autoantigen itself and as a biomarker to monitor patients' response to SSC therapy. The topic is relevant to the field, not only for SSc but in other autoimmune diseases. Likewise, it is an original topic as this review concentrates only on CXCL4, which participates in several pathogenic mechanisms of the disease, offering a wider view than other reviews on the field that mention the participation of CXCL4 from individual points of view (e.g., pDC, autoantibodies origin, dendritic cells).  Being a narrative review, no specific gap in the field is covered, but it may help to provide a new perspective to other studies in the field of SSc. Likewise, being a narrative review, a specific methodology was not followed, although the authors could provide more information about studies that found contradictory results and discuss them; the conclusions are a summary of the main points reviewed.

We thank the reviewer for the positive evaluation of our manuscript.

Regarding the references, they are appropriate for this kind of manuscript

The manuscript includes three figures appropriate for the information being reviewed, although its quality could be improved.

Some very minor points should be addressed.

The title of the submission is different from the title of the manuscript. Please use the title of the manuscript, since the title of the submission do not appropiately reflect the focus of the manuscript.

We apologize, there was a mistake during the submission.

Some minor issues in punctuation need correction (e.g. point at beginning of paragraph, page 6 line 9). 

We thank the reviewer, we have corrected and controlled all text.

Correct "as autoreactive T-cells and autoantibodies play as crucial role in the disease" (page 8).

We thank, we have corrected

On page 3, line 8, it seems that the phase was left unfinished just before reference [6].

We thank the reviewer for noticing this, we have corrected.